# Action-modulated midbrain dopamine activity arises from distributed control policies

**Jack Lindsey**
Department of Neuroscience
Columbia University
New York, NY
jackwlindsey@gmail.com

**Ashok Litwin-Kumar**
Department of Neuroscience
Columbia University
New York, NY
a.litwin-kumar@columbia.edu

## Abstract

Animal behavior is driven by multiple brain regions working in parallel with distinct control policies. We present a biologically plausible model of off-policy reinforcement learning in the basal ganglia, which enables learning in such an architecture. The model accounts for action-related modulation of dopamine activity that is not captured by previous models that implement on-policy algorithms. In particular, the model predicts that dopamine activity signals a combination of reward prediction error (as in classic models) and "action surprise," a measure of how unexpected an action is relative to the basal ganglia's current policy. In the presence of the action surprise term, the model implements an approximate form of $Q$-learning. On benchmark navigation and reaching tasks, we show empirically that this model is capable of learning from data driven completely or in part by other policies (e.g. from other brain regions). By contrast, models without the action surprise term suffer in the presence of additional policies, and are incapable of learning at all from behavior that is completely externally driven. The model provides a computational account for numerous experimental findings about dopamine activity that cannot be explained by classic models of reinforcement learning in the basal ganglia. These include differing levels of action surprise signals in dorsal and ventral striatum, decreasing amounts of movement-modulated dopamine activity with practice, and representations of action initiation and kinematics in dopamine activity. It also provides further predictions that can be tested with recordings of striatal dopamine activity. [1]

## 1 Introduction

An extensive body of work has sought to account for the function of the basal ganglia using the computational framework of reinforcement learning (RL), in particular RL algorithms for action selection and value learning [30, 44, 29, 58]. Striatal neurons within the basal ganglia integrate diverse inputs, including projections from across the cerebral cortex. The activity of these neurons, in particular neurons in the dorsal striatum, plays a key role in action selection [46]. On the other hand, neurons in the ventral striatum have been shown to encode the learned value of stimuli [10, 43, 45]. The phasic activity of midbrain dopamine neurons projecting to the striatum gates plasticity at cortico-striatal synapses [63, 9, 11] and may therefore modulate the learning of actions and values. Indeed, in many RL algorithms, the learning of state-to-action mappings ("policies") and value estimates is modulated by a scalar factor known as the "advantage" or "reward prediction error" (RPE), which measures deviations in attained reward from expectations based on learned value estimates. Numerous experiments have shown that striatal dopamine activity encodes an RPE-like signal [50, 43, 26].

---

[1]Code for our experiments is provided at https://github.com/jlindsey15/ActionDopamineDistributedControl

36th Conference on Neural Information Processing Systems (NeurIPS 2022).

Collectively, these findings suggest a model in which the basal ganglia implements an online actor-critic RL algorithm, where ventral and dorsal striatal subregions play the roles of critic and actor, respectively, and dopaminergic activity encodes the advantage (RPE) signal [30]. This model has been extended in a variety of ways to incorporate more biological detail [56, 57, 11, 8, 55], ideas from model-based RL [15], distributional RL [13], and meta-RL [61].

Despite these promising links, there remain challenges to the view of basal ganglia as implementing an actor-critic RL algorithm. In this work, we address two such challenges. First, dopamine activity in the basal ganglia is observed to encode other information beside RPEs [18], particularly signals relating to movement initiation and vigor [12, 67]. Second, the classic actor-critic model of the basal ganglia is an on-policy RL algorithm—it is designed to learn from experiences driven by the basal ganglia's policy. However, motor control in the brain is distributed, with multiple brain regions including the motor cortex and cerebellum exerting influence on behavior [19, 64, 3, 52, 6]. We show that classic actor-critic models of the basal ganglia fail in this scenario and argue that the ability to learn from off-policy experience is essential to models of basal ganglia learning.

We present a biologically plausible model of off-policy RL in continuous action spaces. Our model differs from the classic actor-critic model of the basal ganglia by adding an additional action-related term to dopamine activity. We call this term "action surprise," as it measures deviation of the action an agent takes in a particular state from the typical action that the output of the basal ganglia would select in that state. We show mathematically that, with this addition, the algorithm implements an approximate form of $Q$-learning in continuous action spaces proposed by [23]. Using simulations, we show that action surprise is essential to effective learning when controllers other than the basal ganglia also contribute to behavior. Thus, action-related activity need not be understood as an independent function of midbrain dopamine neurons separate from their role in learning, but rather as a necessary component of an algorithm that enables off-policy RL. This action surprise model accounts for several experimental findings about movement-related dopamine activity in the basal ganglia. Finally, we describe predictions of our model regarding dopamine activity.

## 2 Background: Connection between policy gradient algorithms and cortico-striatal plasticity

Here we briefly introduce the connection between on-policy policy gradient algorithms and synaptic plasticity rules in the striatum. In what follows, we assume familiarity with standard notation and concepts in RL; for a brief review, see Appendix A.1. Policy gradient algorithms update the parameters $\boldsymbol{\theta}_P$ of the policy according to

$$\Delta\boldsymbol{\theta}_P \propto \nabla_{\boldsymbol{\theta}_P} \log \pi(\mathbf{a}_t|\mathbf{s}_t)\delta_t, \tag{1}$$

where $\delta_t$ is an estimate of the advantage function $A(\mathbf{s}_t, \mathbf{a}_t) = Q(\mathbf{s}_t, \mathbf{a}_t) - V(\mathbf{s}_t)$.

Throughout this work we assume that policies $\pi(\mathbf{a}|\mathbf{s})$ are parameterized by Gaussian distributions $a_i \sim \mathcal{N}(\mu_i(\mathbf{s}), \sigma_i^2)$. For simplicity, we will assume $\sigma_i = \sigma$ for all $i$; however, our results are easily extended to the case of heterogeneous $\sigma_i$. Letting $\boldsymbol{\theta}_{\boldsymbol{\mu}}$ be the parameters of $\boldsymbol{\mu}(\mathbf{s})$, we have

$$\Delta\boldsymbol{\theta}_{\boldsymbol{\mu}} \propto \frac{1}{\sigma^2}\delta_t \left(\mathbf{a}_t - \boldsymbol{\mu}(\mathbf{s}_t)\right)\nabla_{\boldsymbol{\theta}_{\boldsymbol{\mu}}}\boldsymbol{\mu}(\mathbf{s}_t). \tag{2}$$

Supposing our policy is parameterized by a linear map from a feature representation of $\mathbf{s}_t$, $\boldsymbol{\mu}(\mathbf{s}_t) = \mathbf{W}_{\boldsymbol{\mu}}\boldsymbol{\phi}(\mathbf{s}_t)$, this becomes:

$$\Delta\mathbf{W}_{\boldsymbol{\mu}} \propto \frac{1}{\sigma^2}\delta_t \left(\mathbf{a}_t - \boldsymbol{\mu}(\mathbf{s}_t)\right)\boldsymbol{\phi}(\mathbf{s}_t)^\top. \tag{3}$$

This is a "three-factor" learning rule [20] for synapse $W_{ij}$ obtained by multiplying presynaptic activity $\phi_j(\mathbf{s}_t)$, a postsynaptic term $(\mathbf{a}_t)_i - \mu_i(\mathbf{s}_t)$, measuring the deviation of the sampled action from the typical action in this state, and a third factor $\delta_t$. RL models of the basal ganglia assume that $(\mathbf{a}_t)_i - \mu_i(\mathbf{s}_t)$ is available to the postsynaptic neuron and that $\delta_t$ is signaled by dopamine release in the striatum [58, 44]. Experiments have indeed shown that a coincidence of dopamine release and pre and post-synaptic neural activity enables plasticity at cortico-striatal synapses [63, 9, 11]. Biological implementations of this learning rule are discussed further in Appendix A.2.

In actor-critic models the $\delta_t$ factor is

$$\delta_t = r_{t+1} + \gamma\hat{V}(\mathbf{s}_{t+1}) - \hat{V}(\mathbf{s}_t), \tag{4}$$

often referred to as "reward prediction error" (RPE). Here, $\hat{V}$ is an estimate of the value function output by a "critic" network, separate from the policy network, which learns its parameters $\boldsymbol{\theta}_V$ using temporal difference (TD) learning:

$$\Delta\boldsymbol{\theta}_V \propto \delta_t \nabla_{\boldsymbol{\theta}_V} \hat{V}(\mathbf{s}_t). \tag{5}$$

In models of the basal ganglia, the ventral striatum is often assigned the role of the critic, as it is implicated in value-learning but less so in controlling actions [43, 44]. Importantly, TD learning uses the same quantity $\delta_t$ for learning the value function as for policy learning. Hence, a scalar $\delta_t$ signal measuring RPE and broadcast across the striatum supports learning for both the actor and critic. RPE captures key experimental features of striatal dopamine activity—responses to unexpected reward or reward-predictive cues, no significant response to expected reward, and suppression in response to unexpected lack of reward [50, 43, 26].

## 3 Off-policy RL through action surprise signals in dopamine activity

The actor-critic algorithm described above is an on-policy algorithm. Hence, the algorithm may perform suboptimally, or fail altogether, if the actions used for learning are not (always) sampled from the learned policy. Off-policy algorithms are often used to enable learning from a replay buffer of past experiences, expert demonstrations, and/or a separate exploration policy [2]. In a biological context, we argue that the fact that other brain regions can exert control over behavior independently of the basal ganglia further motivates off-policy RL. Indeed, we will show empirically that standard on-policy algorithms suffer when other controllers exert (partial) control over an agent's behavior.

### 3.1 Action-sensitive dopamine activity arises from parameterized $Q$-learning

While there are a variety of approaches to off-policy RL, many require learning an estimate of the $Q$-function $Q(\mathbf{s}, \mathbf{a})$ rather than $V(\mathbf{s})$. $Q$-learning iteratively minimizes the following loss function:

$$\mathcal{L} = \left\| y_{t+1} - \hat{Q}(\mathbf{s}_t, \mathbf{a}_t) \right\|^2, \tag{6}$$

$$y_{t+1} = r_{t+1} + \gamma \max_{\mathbf{a}} \hat{Q}(\mathbf{s}_{t+1}, \mathbf{a}). \tag{7}$$

Computing the quantity $\max_{\mathbf{a}} Q(\mathbf{s}, \mathbf{a})$ directly is intractable in high-dimensional, continuous action spaces. A variety of approaches to this problem have been proposed. Here we focus on an approach adopted by [23], which involves restricting the form of the $Q$-function estimate to a family of functions whose maximum is easy to compute (we discuss other techniques for continuous Q-learning and off-policy RL, which we argue are less biologically realistic, in Appendix A.3).

We parameterize the $Q$-function as follows:

$$\hat{Q}(\mathbf{s}, \mathbf{a}) = \hat{V}(\mathbf{s}) - \frac{1}{\sigma^2} \left\| \mathbf{a} - \boldsymbol{\mu}(\mathbf{s}) \right\|^2. \tag{8}$$

For now we treat the the scaling factor $\sigma$ as a fixed hyperparameter; however, it can also be learned online (see Appendix A.4). This parameterization is a special case of the one proposed in [23].

A primary insight of our work is the observation that, under the parameterization of Eq. 8, gradient updates of the loss function of Eq. 6 yield a biologically plausible actor-critic algorithm with action-sensitive dopamine activity. In particular, taking the gradient of the $Q$-learning loss function with respect to the parameters $\boldsymbol{\theta}_V$ and $\boldsymbol{\theta}_{\boldsymbol{\mu}}$ of $\hat{V}$ and $\boldsymbol{\mu}$, respectively, yields the following learning updates:

$$\Delta\boldsymbol{\theta}_V \propto \delta_t^+ \nabla_{\boldsymbol{\theta}_V} \hat{V}(\mathbf{s}_t), \tag{9}$$

$$\Delta\boldsymbol{\theta}_{\boldsymbol{\mu}} \propto \frac{1}{\sigma^2} \delta_t^+ (\mathbf{a}_t - \boldsymbol{\mu}(\mathbf{s}_t)) \nabla_{\boldsymbol{\theta}_{\boldsymbol{\mu}}} \boldsymbol{\mu}(\mathbf{s}_t), \tag{10}$$

where

$$\delta_t^+ = r_{t+1} + \gamma \hat{V}(s_{t+1}) - \left[ \hat{V}(\mathbf{s}_t) - \frac{1}{\sigma^2} \left\| \mathbf{a}_t - \boldsymbol{\mu}(\mathbf{s}_t) \right\|^2 \right] \tag{11}$$

$$= \delta_t + \frac{1}{\sigma^2} \left\| \mathbf{a}_t - \boldsymbol{\mu}(\mathbf{s}_t) \right\|^2. \tag{12}$$

These update equations are the same as those of the on-policy advantage actor-critic algorithm, but with one additional term added to the dopaminergic signal. Now this signal $\delta^+$ represents a the sum of classic RPE $\delta$ and $\|\mathbf{a} - \boldsymbol{\mu}(\mathbf{s})\|^2$, which measures the deviation of the sampled action from the action most likely to be chosen by the actor. We call this term "action surprise." It can be interpreted as a reduction in the predicted value $Q(\mathbf{s}_t, \mathbf{a}_t)$ of actions that deviate from the basal ganglia's policy.

We note that our model is agnostic to whether action surprise is encoded by the same neurons as RPE. It may be encoded by separate neurons as long as they release dopamine in the same areas as RPE-signaling dopamine. We also note that the update equation (Eq. 9) for the actor uses the action $\mathbf{a}_t$ taken by the agent. Biologically, this requires an efferent copy of the agent's action (taking into account the influence of other controllers) be sent to the striatal projection neurons representing action in the basal ganglia. This architecture is consistent with the presence of pathways from motor cortex, thalamus, and cerebellum to striatal projection neurons [37, 25, 5, 36]. A schematic of the connections involved in our model is depicted in Fig. 1.

It is easy to misinterpret the effect of the dopaminergic action surprise as encouraging the basal ganglia's policy to imitate those of other controllers. This is in fact not the case. Later, we show empirically that this model robustly enables off-policy learning even in the presence of poorly performing external controllers. Below, however, we show how imitation learning can be implemented.

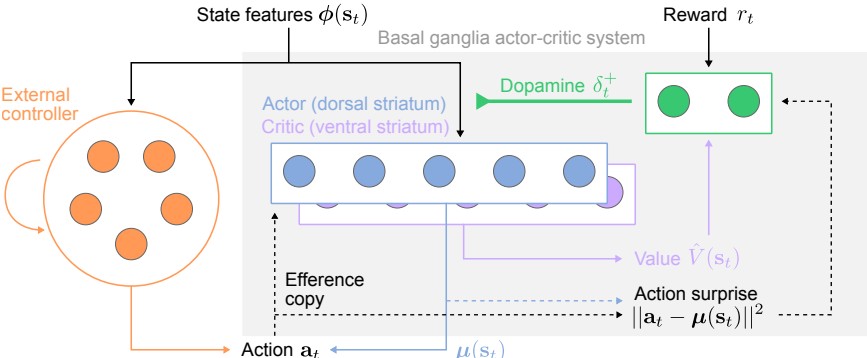

Figure 1: Schematic of model architecture. Actions are driven in parallel by an actor-critic architecture in the basal ganglia (gray region) and an external controller, representing other brain regions (orange). Both systems receive state information via a feature representation $\phi(\mathbf{s}_t)$ (e.g. cortico-striatal projection neuron activity). The actor-critic module performs RL with weight updates modulated by $\delta_t^+$, which is a combination of both RPE and action surprise. Dashed lines indicate architectural features of the action surprise model that are not present in the classic actor-critic model.

## 3.2 Differential action surprise signals in dorsal and ventral striatum

The formulation above predicts that the same dopamine signal $\delta^+$ is broadcast to both the actor and critic. However, the ratio of movement-related to reward-related modulation of dopamine activity appears to vary across the striatum, with more movement-related activity in dorsal regions [27]. In our modeling framework, this corresponds to the action surprise term being weighted more strongly in the actor (dorsal striatum) than the critic (ventral striatum). Such an asymmetry is produced by adding an additional term to the weight updates of the actor, proportional to

$$\|\mathbf{a}_t - \boldsymbol{\mu}(\mathbf{s}_t)\|^2 \, (\mathbf{a}_t - \boldsymbol{\mu}(\mathbf{s}_t)) \nabla_{\boldsymbol{\theta}_{\boldsymbol{\mu}}} \boldsymbol{\mu}(\mathbf{s}_t). \tag{13}$$

This update is aligned with the gradient with respect to $\boldsymbol{\theta}_{\boldsymbol{\mu}}$ of the following loss function:

$$\mathcal{L}_{\text{sup}} = \mathbb{E}\left[\|\mathbf{a}_t - \boldsymbol{\mu}(\mathbf{s}_t)\|^2\right], \tag{14}$$

with the learning rate itself governed by the magnitude of action surprise. Hence, these updates encourage the basal ganglia policy to imitate the agent's behavioral policy on steps with large action surprise. Heuristically, steps with large action surprise are more likely to have been driven by an external controller. An additional action surprise contribution to actor-modulating dopamine activity

can therefore be interpreted as an approximate supervised learning signal encouraging imitation of the external controller's policy. We show in simulations that this effect helps the basal ganglia more rapidly consolidate expert policies of other controllers.

# 4 Experimental Setup

*Tasks*. We simulated two simple continuous control tasks to demonstrate the role of the action surprise term in off-policy learning. See Fig. 2A for illustrations.

*Open-field navigation:* An agent is positioned in a two-dimensional square environment with continuous coordinates. On each trial, the initial position of the agent and a goal location are randomly sampled. The agent controls its $x$ and $y$ acceleration.

*Two-joint:* An agent controls a two-joint arm with two equal-length segments. The arm position and a target location are randomly sampled at the beginning of each trial. The agent outputs torques at each of the joints in order to move the most peripheral point of the arm toward the target.

In both tasks the cost incurred by the agent at each time step is proportional to the inverse of its squared distance from the goal location $\mathbf{g}$ at each time step, plus penalties for its squared velocity and acceleration (summed across joints in the two-joint arm case)

$$r_t = -\left( \|\mathbf{x}(t) - \mathbf{g}\|^2 + \alpha_1 \left\| \frac{d\mathbf{x}(t)}{dt} \right\|^2 + \alpha_2 \left\| \frac{d^2\mathbf{x}(t)}{dt^2} \right\|^2 \right) \tag{15}$$

We also experimented with a version of the loss function in which rewards for reaching the goal location are sparse and binary, rather than continuous, which gave similar results (see Appendix A.7).

***Model architecture***. For both tasks we used a neural network architecture with a single hidden layer. The state inputs to the network consisted of the agent's (angular) position and (angular) velocity, as well as the position of the target location, a total of six scalar variables. Each variable was represented as a one-hot vector by discretizing its domain into 10 equally spaced bins. These vectors were concatenated and used as the network input. The hidden layer had size 256 and used ReLU nonlinearities. Input weights to the hidden layer were fixed at random (Kaiming uniform) initializations. We fixed these weights to avoid the complexity of biological implementations of backpropagation, since this is not the focus of our work (however, we note that the action surprise model can be applied to deep networks by backpropagating gradients through additional layers). Thus, the hidden layer activations served as a fixed feature representation $\phi(\mathbf{s})$ of the environment state, and all learning occurs in output weights $\mathbf{W}_{\mu}$ and $\mathbf{w}_V$ which output actions $\boldsymbol{\mu} = \mathbf{W}_{\mu}\phi(\mathbf{s})$ and value estimates $\hat{V} = \mathbf{w}_V \cdot \phi(\mathbf{s})$, respectively. We refer to this network as the basal ganglia network.

***External controllers and training protocol***. To model the influence of other brain regions on behavior, we introduced an additional neural network that also exerted control over the agent's actions (Fig. 1, orange). To vary the performance of this controller's policy, we trained the controller on the task with backpropagation for different numbers of steps: 0 (random policy), 2,000 (intermediate policy), or 100,000 (expert policy). We also varied the degree to which the external controller influences behavior. In the fully on-policy case, the external controller is not used. In the fully off-policy case, the external controller entirely drives the behavior. In the case of partial control, each action is sampled from the basal ganglia's network or the external controller with probability 0.5 each. Additionally, we implemented an alternative partial control mechanism, in which the average of the basal ganglia network output and the external controller output is used.

Throughout training, Gaussian exploration noise is added to the output of both networks at each step. In all cases, for the action surprise model, the action $\mathbf{a}_t$ used for the actor update in Eq. 9 is the action taken by the agent, taking into account the contributions of both controllers and the exploration noise.

***On-policy baseline models***. We compared the action surprise model with two baselines in which dopamine neurons signal pure RPE. The first is described by Eqs. 2 and 5, where the action used the actor update is simply the action $\mathbf{a}_t$ taken by the agent, as in the action surprise model. As discussed above, this approach is potentially unstable in the presence of other controllers due to the mismatch between the behavioral policy from which actions are sampled and the basal ganglia network's policy. As a second, potentially more competitive baseline, we instead used the output of the basal ganglia network alone, including exploration noise, in place of $\mathbf{a}_t$ in Eq. 2. We refer to this as the "no efferent

copy" model since the basal ganglia is blind to the action ultimately taken by the agent. In this version, the influence of the external controller is effectively treated as part of the environment. We note that this approach has no ability even in principle to learn from fully off-policy data, but can learn when the basal ganglia network exerts partial control over actions.

*Hyperparameters*. For all models we optimized the learning rate and magnitude of exploration noise as hyperparameters. For the action surprise model we optimized the coefficient $\frac{1}{\sigma^2}$ of the action surprise term as a hyperparameter. To ensure strong baselines we allowed the actor and critic learning rates to be optimized separately for the RPE-only models. See Appendix A.5 for details.

# 5  Experimental Results

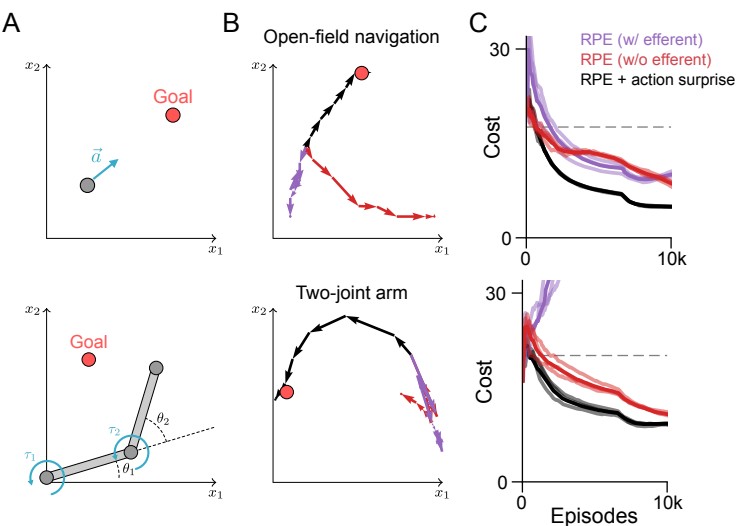

Figure 2: **A**: Depictions of simulated tasks. **B**: Example trials in which the action surprise model successfully reaches the target but the RPE-only models fail to. **C**: Performance of the three models when the basal ganglia shares control of behavior with an external controller. Dark lines indicate mean performance over three runs, and faint individual lines (in this case difficult to see due to tight overlap) indicate individual runs. Performance shown reflects policy of the basal ganglia network and external controller. Dashed line indicates the performance of the external controller alone.

We first analyzed performance for the case of actions partially driven by the basal ganglia network and partially driven by the external controller, intended as a representative model of behavioral control distributed across brain regions. Fig. 2C shows that the basal ganglia policy attained superior learning speed and performance than RPE-only models (shown here is a representative case using the intermediate-level external controller). Fig. 2B shows example trials in which this performance advantage manifests very clearly, with the action surprise model successfully reaching the target while the other models fail completely. The action surprise model remained advantageous when we varied the expertise of the external controller (Fig. 3A). The same was true when we assessed combined performance of the basal ganglia network and the external controller (Fig. 3B), rather than performance of the basal ganglia network by itself. Notably, the RPE-only model with efferent copy failed to learn in some cases. We found similar results when the contributions of the basal ganglia network and external controller were averaged rather than combined by sampling (Appendix A.6).

We next examined the fully off-policy case. The RPE-only model with no efferent copy failed to learn from off-policy data in all cases, as expected, as the action term used in the update for this model is uncorrelated with the actions taken by the external controller. Interestingly, the RPE-only model with efferent copy also failed (catastrophically) to learn in all cases. The action surprise model, by contrast, learned both tasks regardless of the quality of the external controller's policy (Fig. 3C).

A potential concern is that the action surprise model's improvement for off-policy learning comes at the expense of on-policy learning performance. However, we found that even in the case of fully

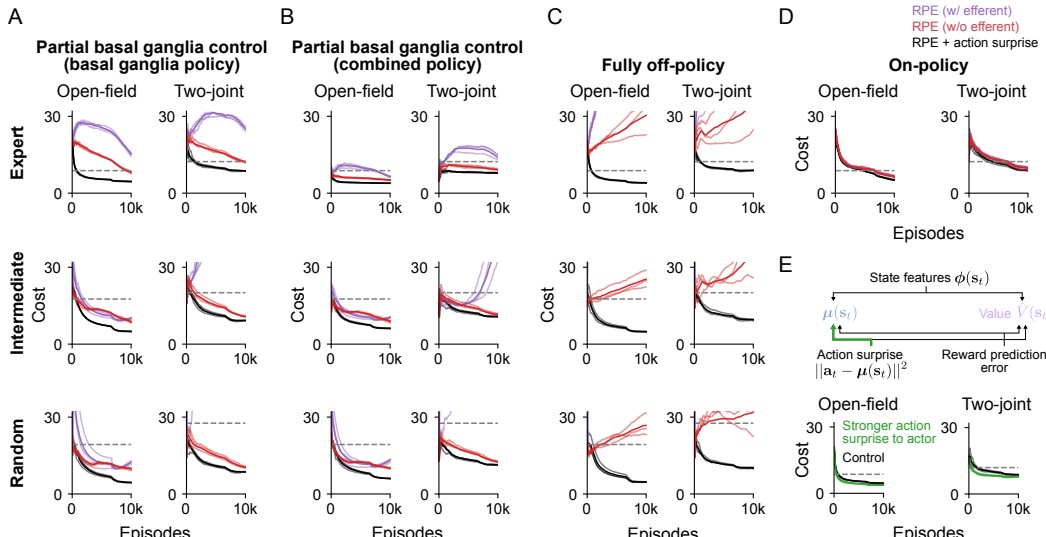

Figure 3: **A**: Performance of the three models, as in Fig. 2C, shown for external controllers with varying levels of expertise (random, intermediate, expert; trained via backpropagation on the task for different numbers of steps). Middle row corresponds to Fig. 2C. **B**: Same as A, but showing the performance of the combined policy of the basal ganglia network and external controller. **C**: Same as A, but in the case where all behavior is driven by the external controller during learning. **D**: Performance of the RPE and RPE+action surprise models for fully on-policy learning on the two tasks. **E**: Top: Schematic of model in which the contribution of the action surprise term to dopamine activity is weighted more strongly in the actor than in the critic (green arrow). Bottom: comparison of this model to the original action surprise model in the case of partial basal ganglia control and fully off-policy learning with the expert controller (black traces are the same as in the top row of panel E). In this example the action surprise term is weighted 8 times as strongly in the actor than the critic.

on-policy learning, with no external controller, the action surprise model matched and indeed slightly outperformed the RPE-only model (Fig. 3D).

We also tested the variant of the action surprise model in which an additional update (Eq. 13) is applied to the actor, corresponding to a higher action surprise coefficient in the actor (dorsal striatum) than in the critic (ventral striatum). We found that this addition improves learning in the presence of expert external controllers (Fig. 3E), consistent with accelerated consolidation of the expert policy.

## 6 Biological implications of the action surprise model

The action surprise model explains several features of midbrain dopamine activity and also makes several testable predictions, which we outline below.

***Nonspecific encoding of action***. A central property of the model is that the action surprise term $\|\mathbf{a}_t - \boldsymbol{\mu}(\mathbf{s}_t)\|^2$ is not action-specific—it reflects only scalar information about the agent's action, even in high-dimensional action spaces, and does not distinguish between two equally surprising actions. This contrasts with the representation in striatal projection neurons forming the output of the actor network, which specify movement commands. Indeed, experimental recordings have found detailed encoding of kinematics in striatal projection neurons [16] but only coarse movement-related signals in dopamine activity that do not reliably distinguish between movement types [12, 38]. This distinction is consistent with our model and inconsistent with models that explain action-modulated dopamine activity in terms of specific motor commands.

***Decrease in action-modulated dopamine activity with learning***. The action surprise model predicts that movement-related dopamine activity is lower when the basal ganglia's policy more closely matches the agent's actions. Before the basal ganglia has learned an effective policy, actions driven by an expert external controller will typically provoke large action surprise. Once the basal ganglia

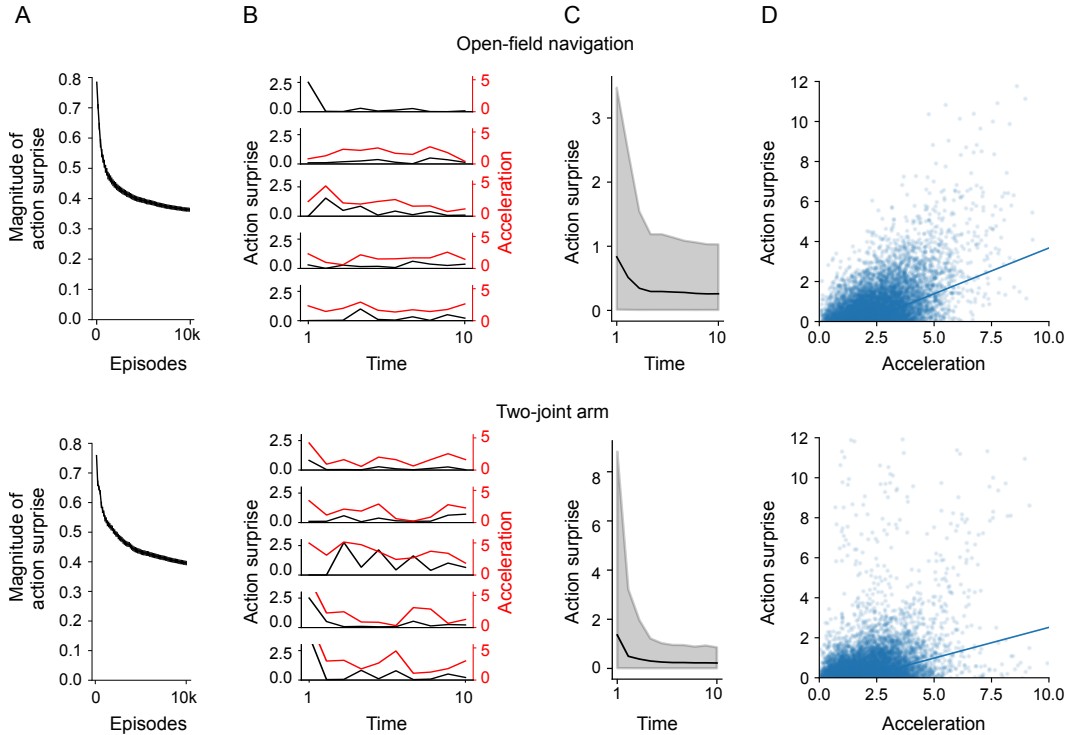

Figure 4: **A**: Mean and standard deviation of action surprise term in the dopamine response over the course of learning, in the case where the basal ganglia shares partial control with an expert controller. In all these examples $\frac{1}{\sigma^2}$ was set to 0.125 and and the exploration noise to 1.0. **B**: Traces of the action surprise signal (black) and the the agent's acceleration (red) – or rotational acceleration averaged across joints in the two-joint arm case – for five randomly sampled example trials. **C**: Mean and 95% confidence interval (across trials) of action surprise as a function of the time within a trial. **D**: Scatter plots (showing individual steps, across many episodes) of the action surprise term in the model dopamine response vs. magnitude of agent acceleration.

network has learned an expert policy itself, action surprise will typically be lower. Thus, we predict a reduction in the magnitude of action surprise signals over the course of learning. We observed this phenomenon in both simulated tasks (Fig. 4A). This prediction is supported by findings that dopamine activity coinciding with lever-press movements decreases with repeated task practice [12, 22].

***Context-dependent correlation between dopamine activity and acceleration***. Several studies have shown that striatal dopamine activity correlates with movement speed and acceleration [60, 49, 4]. Others [27, 12, 38] have shown that activity in movement-responsive midbrain dopamine neurons is modulated primarily movement at initiation (and in some cases at movement offset) but less so during ongoing movements. Our model gives rise to a complex relationship between movement and action-driven dopamine signals, with the two appearing correlated on some trials but not others (Fig. 4B). On average, action-driven dopamine activity is highest at trial onset (Fig. 4C), and it is positively correlated with the agent's acceleration (Fig. 4D). Our results suggest that the representations of movement initiation and kinematic information attributed to dopamine activity in prior work may actually emerge as byproducts of an action surprise computation.

***Neural circuits underlying the computation of action surprise***. The action surprise model requires information about the agent's action be available to midbrain dopamine neurons for computation of the action surprise term. This is consistent with the presence of pathways from motor cortex and cerebellum to midbrain dopamine neurons [62, 39]. It is also consistent with perturbation experiments finding a causal effect of cortical perturbations and lesions on striatal dopamine activity [33, 54, 1, 7]. We note that the specific form of the action surprise term in our model, $\|\mathbf{a} - \boldsymbol{\mu}(\mathbf{s})\|^2$, is somewhat arbitrary, and other measures of distance between $\mathbf{a}$ and $\boldsymbol{\mu}(\mathbf{s})$ may be equally suitable. We also note that our model is agnostic as to whether the action surprise and RPE components of dopamine activity

are represented in the same individual neurons. Empirically, RPE-signalling and movement-signaling dopamine neurons appear to comprise distinct but partially overlapping populations in the same striatal subregions [12], though data on this question is limited.

***Considerations for future experimental work***. We intend this work to motivate experimental work that can support, challenge, or refine our model. Doing so requires careful experimental design. First, the effects of action-related dopamine activity on learning are easily overlooked in the context of simple tasks in which animals learn to associate cues or actions with immediate reward. In such tasks, standard RPE-based models may be adequate to achieve good performance. Second, disentangling "action surprise" as we define it from other related quantities (initiation, acceleration, motivation, etc.) requires a rich action space, while many animal RL studies involve a small, discrete set of actions (e.g. binary choice). Hence, motor tasks that require sequences of movements involving continuous control [31, 41] are most appropriate for our purposes. Testing predictions of the model also involves disentangling the behavioral contributions of the basal ganglia and other brain regions and analyzing these in concert with simultaneously recorded dopamine activity and behavior. While the technology exists to conduct such simultaneous recordings, no such experiment has been performed. Tests of the model would also benefit from the ability to causally modulate the surprisingness of actions, for instance using catch trials with different target movements.

## 7 Discussion

Our work demonstrates a new link between action-related midbrain dopamine activity and reinforcement learning in the basal ganglia. While each of these topics has received extensive treatment in the neuroscience literature, they have typically been studied separately. Midbrain dopamine activity is known to causally affect movement initiation and invigoration in real time [12, 47]. Consequently, the function of such activity is often regarded as a motivational signal [65, 42], or as a contributor to action selection [17], separate from dopamine's role in reinforcement learning. However, action and RPE-related dopamine release coincide in the same striatal subregions, often even in the same neurons, and with comparable magnitude [60, 49, 12, 38, 18]. Thus, it is likely that both action-related and RPE-related dopamine activity impact cortico-striatal synaptic plasticity and learning. Excitingly, concurrent experimental work provides direct evidence for an action surprise-like signal in dopamine neurons projecting to the tail of the striatum, and for a causal influence of these neurons on task learning [22]. Our results show that such an influence is a necessary component of an architecture capable of off-policy learning.

We note that our proposal does not preclude alternative roles for dopamine activity. Indeed, action-related dopamine signals are observed to both precede and follow movement [12]. Early responses may participate in motivation and action selection while lingering responses responses (extending into the critical plasticity window for cortico-striatal synapses [66]) may aid in learning. Whether similar signals can achieve both functions is an important direction for future modeling work. Prior work [17] has suggested that dopaminergic action selection signals may correspond to imagined RPEs of simulated action outcomes according to a learned model of the environment, which may correlate strongly with action surprise if actions are partially controlled by a model-based system.

Our model suggests that distributed control of behavior by many regions plays an important role in learning in the basal ganglia. A number of experiments have observed differential contributions from multiple brain regions during learning. For instance, some motor skills are observed to recruit the motor cortex early in learning before being consolidated into the basal ganglia [31, 28]. These results suggest that regions other than the basal ganglia may be more adept at flexibly adapting to novel tasks, while the basal ganglia specializes in consolidating well-practiced skills. Other studies have observed parallel contributions of model-based and model-free reinforcement learning strategies in the same task [53, 21], revealing arbitration mechanisms leverage the advantages of each [34, 32]. The flexibility afforded by off-policy RL algorithms enables the basal ganglia to benefit from complementary learning and control strategies adopted by other neural circuits. Exploring how off-policy RL algorithms can best leverage these diverse sources of expertise is a fruitful avenue for extensions to our model. For instance, action surprise signals may be modulated by the confidence of external controllers in order to more efficiently learn from expert behavior.

Our work also has implications for reinforcement learning. The introduction of a biologically plausible off-policy reinforcement learning algorithm, involving local learning rules and a single

modulatory factor, enables the deployment of off-policy RL on plasticity-enabled neuromorphic hardware [14]. Moreover, we anticipate that insights into how neural circuits learn from off-policy behavior governed by distributed and diverse controllers will provide inspiration for RL algorithms.

## Acknowledgments and Disclosure of Funding

This work was supported by NSF NeuroNex Award DBI–1707398 and The Gatsby Foundation (GAT3708). ALK was also supported by the McKnight, Burroughs-Wellcome, and Mathers Foundations. JL was also supported by the DOE CSGF (DE–SC0020347). The authors declare no competing interests.

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
