# OpenReview forum: "Action-modulated midbrain dopamine activity arises from distributed control policies"
_NeurIPS.cc/2022/Conference — NeurIPS 2022 Accept_

### Official Review · Reviewer_15y6 · 2022-06-20

**Rating:** 9
**Confidence:** 5
**Soundness:** 4 excellent
**Presentation:** 4 excellent
**Contribution:** 4 excellent

**Summary:**

the currently dominant view of RL in the basal ganglia. The authors then show how Q-learning with a particular state-action value function parametrization yields an RL algorithm that is very close to the actor-critic model while being able to learn off-policy. Importantly, this new model has an "action-surprise" term that also allows the model to explain the existence of movement-related modulation in the striatum. The authors show that the model can produce performance advantages. Finally, they discuss how the model might be implemented biologically.

**Questions:**

See comments above.

**Limitations:**

The authors don't really address limitations of their approach. I don't see any issues with negative societal impact.

**Strengths And Weaknesses:**

Overall, I really liked this paper. It presented a novel theoretical idea that jointly solves both empirical and computational problems. It was clearly written and compellingly argued.

My only major comment is that I wish the authors had simulated actual neuroscience experiments rather than (or in addition to) their toy setups (the two-joint arm and open field). This feels like a missed opportunity to show how the model can directly explain the relevant empirical data.

Related to the last point, there are experiments that measure dopamine when other action-related areas are manipulated (e.g., inactivated, stimulated, lesioned). It seems like this model should make predictions about these experiments. Similarly, if an animal's actions are influenced in unexpected ways, then the prediction is that this should register in the dopamine signal (though one would have to be careful to control for sensory effects); I'm not aware of studies that look specifically at this question, but it seems like a clear prediction.

Minor comments:

- p. 1: "decreasing amounts movement-modulated dopamine activity" -> "decreasing amounts of movement-modulated dopamine activity"

---

> ### Author Response · Authors · 2022-08-02
> **Response to Reviewer 15y6**
>
> We thank the Reviewer for the thoughtful comments.
>
> – We agree that tying our simulations more closely to the experimental neuroscience studies is an important direction for this work and we are actively pursuing this direction.  Many animal RL studies use fairly simple action spaces (e.g. binary choice) which make it difficult to distinguish action surprise from other factors, and thus not suitable for illustrating the properties of our model.  As the Reviewer suggests, an ideal experiment would involve clear variability in the “unexpectedness” of actions, and ideally a way to causally manipulate this factor.  We are working with experimental collaborators to design appropriate experiments and have added a discussion of these experimental considerations to the revised draft of the paper (Section 6, “Considerations for future experimental work”).  The need to control for “sensory surprise” confounds is indeed one of the major challenges in designing an airtight experiment.
>
> – We appreciate the Reviewer’s comment about additional experimental predictions we could make.  There are indeed studies showing that stimulation of motor cortex (and prefrontal cortex) directly affects striatal dopamine release (Kosillo et al. 2016, Strafella et al. 2003, Adrover et al. 2020), and that cortical lesions reduce striatal dopamine release (Boyeson et al. 1985).  We have added these references to the revised draft of the paper (Section 6, “Neural circuits underlying the computation of action surprise”).  To our knowledge there are no studies that clearly distinguish the effects of such manipulations on reward-related and movement-related dopamine activity separately.

---

### Official Review · Reviewer_QgmR · 2022-07-12

**Rating:** 5
**Confidence:** 4
**Soundness:** 2 fair
**Presentation:** 2 fair
**Contribution:** 2 fair

**Summary:**

The authors explains the role of dopamine response at action initiation based on an off-policy actor critic architecture.




**Questions:**

Dopamine neuron's cue response is quite fast, often preceding action execution. The present framework assumes the use of efference copy of finally selected action somewhere in the brain. Does the timing really match up?

Here is a conceptual question. Action value equation (8) assumes any action that deviates from mu(s) are worse. On the other hand, the advantage equation (12) favors action that deviates from mu(s). Where does this apparent inconsistency come from? Maybe it is a matter of whether you apply (8) to V(s(t)) or V(s(t+1)).

Uchibe proposed an off-policy RL scheme with mixture of controllers, as in the present setting, but uses important sampling, which gives more weights on actions similar to one's own; apparently opposite to what is done by action surprise. Could you comment on the relationship or advantages/weakness?
Uchibe E (2018). Cooperative and Competitive Reinforcement and Imitation Learning for a Mixture of Heterogeneous Learning Modules. Front Neurorobot, 12, 61. https://doi.org/10.3389/fnbot.2018.00061



**Limitations:**

The approximation of action value by (8) with fixed sigma would limit the model's applicability.

I updated my score after reading the paper [22].



**Strengths And Weaknesses:**

This paper makes an important point that the brain's action selection is distributed, so that learning should be off-policy.

The dopamine response at action initiation promotes imitation learning of external controller, which works nice if the external controller is doing a fair job, but can screw up the basal ganglia learning if others are doing badly.

Dopamine response at action initiation had been predicted upon finding an high advantage action in a model-based framework (Doya 1999, Figure 7). This should also be discussed.
Doya K (1999). What are the computations of the cerebellum, the basal ganglia, and the cerebral cortex. Neural Networks, 12, 961-974. https://doi.org/10.1016/S0893-6080(99)00046-5

---

> ### Author Response · Authors · 2022-08-02
> **Response to Reviewer QgmR (part 1)**
>
> We thank the Reviewer for the thoughtful comments, which raise a number of useful references and biological considerations.  We respond individually to the questions and comments below.  As a broader point, we would like to emphasize that we view the derivation (and corroborating simulations) of the function of action surprise in off-policy learning as the primary contribution of this work.  Most prior models of dopamine responses (including, for instance, the model in Fig. 7 of the Doya 1999 paper) do not implement off-policy learning algorithms.  The connection between action surprise and off-policy RL provides a strong hypothesis with clear computational motivation about the role of action-modulated dopamine activity in learning, which makes a number of experimentally testable predictions.  We hope these will motivate future experiments.
>
> – Regarding the use of a fixed σ, we show in Appendix A.4 that the value of σ can in fact be learned online with a simple learning rule, and we conduct simulations showing that this procedure is effective.  We are agnostic as to the biological mechanism by which σ may be learned, but the fact that it can be done may alleviate some of the Reviewer’s concerns on this point.
>
> — Regarding the timing of action-related dopamine activity, this is indeed an interesting and important question that deserves more discussion in the paper.  Precise results vary across studies, but dopamine activity is observed to often be broadly tuned around movement onset, with ramping typically preceding onset and residual activity continuing following onset.  Fig 1D of our reference [10] (Da Silva et al., Nature, 2018) provides a good illustration of this.  The relative timing of movement/decision-related striatal activity (including any efferent copy information from other regions) is difficult to pin down, but it is likely that a substantial portion of movement-related dopamine activity follows the corresponding striatal activity.  Whether this picture supports our model depends on the form of the dopamine-modulated plasticity rule at cortico-striatal synapses.  Experiments suggest that dopamine activity  promotes plasticity in a critical window (0.3 to 2s) following striatal activity (Yagishita et al., Science, 2014).  Thus we believe it is likely that the dopamine activity is appropriate for implementing the kind of learning rule we propose.  However, it is indeed true that observed dopamine activity which significantly precedes movement is unlikely to be playing the role our model suggests, and is more likely to be causally involved in movement.   Broad temporal tuning of movement-related dopamine release may allow it to play both these roles at once.  We have added a discussion of these points to the revised version of the paper (Section 7, first paragraph).
>
> — We want to clarify a possible misconception about the relationship between the action surprise model and imitation learning.  While adding action surprise to the actor alone does indeed promote a form of imitation learning (as discussed in section 3.2), its effect is different when present in both the actor and the critic.  As we show in the paper, the effect is to implement a form of Q-learning, which can learn effectively even based on experiences driven by a poorly performing external controller.  To take one example, the bottom row of Fig. 3C shows the action surprise model learning a policy, solely using off-policy experience driven by a low-performing controller, that substantially outperforms that controller.  Thus, the basic action surprise model is more robust to the specifics of the other controller’s policy than imitation learning (and ultimately this somewhat unintuitive behavior is best understood by thinking of the connection to Q-learning).  We have added a clarification of this point at the end of Section 3.1.
>
> — Regarding the different signs in equation (8) and equation (12), this is indeed because the ||a - μ(s)||^2 term is subtracted from Q(s_t, a_t), but not (because of the max function) from max_a Q(s_{t+1}, a).  Thus in the loss function of equation (7), a negative ||a - μ(s)||^2 term is subtracted, hence the positive sign for action surprise in equation (12).  We have added a clarification of this point following the derivation in Section 3.1 and reformatted equation (11) to reflect this.
>
> (response continued below)

---

> > ### Author Response · Authors · 2022-08-02
> > **Response to Reviewer QgmR (part 2)**
> >
> > — We thank the Reviewer for the reference to the Doya 1999 paper.  As the Reviewer states, this paper proposes a model that involves computing reward prediction errors (advantages) corresponding to simulated outcomes of imagined/planned actions in a model-based framework.  If these advantage estimates are represented by dopamine neurons, it would provide an alternative account of dopamine activity at movement initiation.  In our understanding, this proposal may account for the causal role of dopamine at movement onset, but is less concerned with the effects that movement-related dopamine activity may have on learning.  As such, the Doya model and ours may be considered complementary, rather than competing, in that we seek to account for different phenomena. That being said, if the two models predict very different dopamine response profiles, it would be awkward to reconcile them.  Do they? In the Doya account, dopamine activity signals movements that are likely (according to a learned environment model) to yield outcomes s_{t+1} with high values V(s_{t+1}), but which have not yet been explored frequently enough for this to have been consolidated into the agent’s learned value estimate V(s_t).  In our model, dopamine activity signals movements that are unlikely to be sampled by the basal ganglia policy (high action surprise).  If the other controller in our model is making decisions based on a learned model,  these high-action-surprise actions will tend to be those which are regarded optimistically by the model-based module but have not been explored much by the basal ganglia.  Thus, intuitively, our model and Doya’s may make overlapping predictions in many cases.  We leave it to future work to more formally establish a quantitative connection between these models.  We appreciate the Reviewer pointing out this interesting connection – we have added this reference and a brief discussion of it to the revised paper (Section 7, first paragraph), where we note that the focus of our work is on the impact movement-related dopamine activity on learning, not action selection.
> >
> > – Regarding the importance sampling proposal: as the Reviewer notes, this model weights learning updates more strongly for actions that are likely under the current policy.  This is equivalent to multiplicatively scaling the dopamine factor by (some increasing function of) the likelihood of the action under the current policy.  By contrast, our model involves adding to the dopamine response a term that is inversely related to the likelihood of the action.  While the two approaches do seem contradictory at first glance, they in fact address the same problem, namely that an unmodified reward prediction error response will unfairly assign credit to states from which off-policy actions are taken.  Importance sampling deals with this by effectively ignoring highly off-policy actions.  Our approach deals with this by attempting to account for the change in expected value that arises from taking off-policy actions.  The importance sampling approach will result in slower learning as behavior becomes more and more off-policy, while our approach can still learn efficiently from entirely off-policy data.  The tradeoff is that our approach potentially introduces bias if the assumed form of the Q-function is inaccurate.  We have added a discussion of this point, and the associated reference, to the revised version of the paper (Appendix A.3, with the discussion of other off-policy RL algorithms).

---

> ### Comment · Reviewer_15y6 · 2022-08-07
> **correspondence between review and rating?**
>
> Reviewer QgmR brings up some good points, but as far as I can tell no fundamental criticisms of the paper. So I'm a bit perplexed why the rating is so low. I'd like to advocate that the rating be increased so that the review and rating are commensurate, particularly in light of the authors' detailed responses. Or the reviewer should justify in their review why their rating is so low. Please let me know if I'm missing something.

---

### Official Review · Reviewer_YRYS · 2022-07-14

**Rating:** 6
**Confidence:** 4
**Soundness:** 2 fair
**Presentation:** 3 good
**Contribution:** 3 good

**Summary:**

The paper proposes a new learning rule based on action surprise, which is meant to facilitate off policy reinforcement learning and explain a variety of biological results from DA neuron and basal ganglia recordings, some of which (such as DA coding of novelty and other action related codes) were not thought to be directly linked to reinforcement learning. The authors provide some empirical results comparing the proposed model to other alternatives and show relevant correlates to some biological results, but not direct comparison to behaviour.

**Questions:**

To improve the paper, the authors could do the following:
- provide more examples about the representation of mu(s_t) and how could it be investigated in neural recordings for example (i.e. is there any specific rather than vague/indirect evidence for such representation)
- address the issue of biological/behavioural relevance of tasks and particularly training methods, as they seem more common in machine learning RL, not behavioural RL (although open field tasks are definitely used widely, but not sure if in exactly the same way), paying particularly care for biological realism of timelines, training protocols and learning rules (e.g. is backprop biologically realistic?..)
- perform direct comparison with behavioural learning data or at the very least discuss how exactly it could be done, or if not, why not.
- perform more meaningful comparisons with other models, if authors feel that such comparisons are needed for the paper (it seems in many cases those other models simply don't work, with only general theoretical explanation provided why not).
As a general comment, I'm quite skeptical of the widespread practice of comparing different models in modelling papers, especially when it's used not as a means to select between several alternatives equally considered by authors, but showcase of the proposed model's superior performance to other existing models, as implementation details of such alternative models may not be chosen with the best intention to make them perform as well as possible, and various creative choices (such as best formalisation of states, actions, rewards) and/or fine tuning may be needed for that purpose, so I think absolute assessment and direct comparison with behavioural or physiological data as opposed to other models are much more useful.

Minor issue: why is there a green line in fig. 3E bottom?

**Limitations:**

There was some discussion of limitations, but not really the main limitations/weaknesses of work as I see them (detailed above).

**Strengths And Weaknesses:**

I think the proposed idea is indeed very interesting and has great potential to link a variety of seemingly disparate biological results regarding DA and basal ganglia physiology. I also think the authors perform a great literature review and brilliant integration with relevant findings in literature. The clarity of the proposed model is sufficient, although it's a bit abstract at times, e.g. it would be great to see how the mu(s_t) and a_t - mu(s_t) terms would look in practice, as that's the key element of the model and perhaps not sufficiently explained with examples.

My main problem with the paper is that although it seems to clearly target computational cognitive neuroscience (as opposed to machine learning) audience, the type of tasks, training protocols and measurements seem to correspond a lot more to "machine learning" RL, as opposed to what is or could be done in behavioural experiments, whether animal or human. The type of (distance/error dependent) cost function that is presumably used has little in common with that is used in reinforcement learning, where rewards are often scarce and don't provide constant feedback. Training protocols requiring 100k episodes (or anything of that order) are also totally unrealistic in biology, at least if you interpret this as behavioural trials done in learning (and even if you try to relate it to replay of sequences and similar data, 100k is still quite a stretch). There is no behavioural data provided for these 2 tasks that could suggest based on what timeline or mechanisms animals actually learn in these cases, and comparisons with other models (particularly examples, like in Fig.2b given without much explanation of why this happens) are not very informative. While I don't doubt that authors' model can perform well as asserted and their idea (if not necessarily the implementation) may be a great way to reconcile some disparately looking biological data, I think the effort to link model performance to actual experiments is insufficient. It's true that there is not a great deal of data of direct relevance to off-policy RL in behavioural neuroscience, but I doubt that nothing could be found which could be compared more directly.

---

> ### Author Response · Authors · 2022-08-02
> **Response to Reviewer YRYS**
>
> We thank the Reviewer for the thoughtful comments.  We address the questions and comments below:
>
> — Regarding the large number (100k) of episodes, we have a few comments.  First, the x axis of the performance plots was mislabeled — there were only 10k episodes (each  10 frames, so 100k frames total).  We apologize for the error and have corrected this in the revised version of the paper.  We also note that in most experiments the models reach near-optimal performance much earlier than 10k episodes, after a few thousand episodes.  These timescales are a bit more in line with the number of trials needed to train rodents or primates to perform new motor tasks.  Nevertheless, we do acknowledge that there is a significant gap between the efficiency of these RL algorithms and that of animal learning, and we do not claim that our model has closed this gap entirely.  We suspect that better feature representations of the state information could accelerate learning, as well as the use of qualitatively different algorithms (e.g. model-based) in concert with the model-free approach used here.  Animals may also benefit from a degree of pre-training based on prior life experience that is not available to a model trained from scratch.
>
> — Regarding connections/comparisons to experimental behavioral data and neural recordings, we agree that this is an important and natural direction for this work, which we intend to pursue.  While we are unable to perform in-depth comparisons with specific datasets in the timescale of this reviewing process, we have added comments to the revised version of the paper that specify in more detail our plans to do so (Section 6,  “Considerations for future experimental work”).  First, we note that the predictions of this model are difficult to test without a rich action space — many animal RL studies use fairly simple action spaces (e.g. binary choice) which make it difficult to distinguish action surprise from other factors.  This consideration motivated our choice of tasks with a continuous control component.
>
> Second, testing key predictions of the model involves disentangling the behavioral contributions of the basal ganglia and other brain regions.  While we may be able to get somewhere with purely behavioral data by making educated guesses about which behaviors are BG-driven or not, ultimately recordings of striatal activity (in both medium spiny neurons and dopamine release) are needed to rigorously assess these model predictions.  This is a key future direction for our model.
>
> – Regarding the reward function used in our simulated tasks, we agree that a continuous distance-dependent reward measure is likely not biologically realistic.  We have replicated key experiments shown in Figures 2 and 3 for the case of sparse rewards attained only when the agent reaches a threshold distance from the goal location, and added this as an appendix (Appendix A.7) in the revised version of the paper.
>
> — Regarding comparisons to other models, we agree that model comparisons are often difficult to get right, especially between qualitatively different models with different (hyper)parameters.  We view the derivation of the correspondence between action surprise and Q-learning as the primary contribution of this paper, with the experiments intended to support this finding.  The comparisons performed in this paper are fairly specific, more akin to ablation experiments, and meant to illustrate the importance of the action surprise term in achieving good performance in the off-policy regime, given the basal ganglia-inspired actor-critic architecture used in this paper.  We attempted to be as fair as possible to the action surprise-free baselines, trying two versions (with and without efferent copies of actions) and performing a hyperparameter sweep over actor learning rates, critic learning rates, and exploration noise magnitudes.  Indeed, in our hands the performance with and without action surprise is the same in the on-policy regime, indicating that our implementation of the action surprise-free models is appropriate.
>
> — Regarding the green line in figure 3E, we apologize for the labeling error.  The green line is the model with additional action surprise added to the action module (the black line is the control, with equal levels of action surprise in both the actor and the critic).  We have corrected the labeling in the revised version of the paper.

---

### Meta-Review · Area_Chair_o9ti · 2022-09-13

**Recommendation:** Accept
**Confidence:** Less certain

**Metareview:**

This paper presents an off-policy RL model for providing biologically plausible explanations of of dopamine activity and basal ganglia recordings.

The reviews express a positive evaluation for the research question and framework and the biological plausibility within the context of the domain application. There are some concerns as to whether these models would compare to more well-established behaviour models. The authors have presented robust clairifications of the main concerns of the reviewers.

**Award:**

No

---

### Decision · Program_Chairs · 2022-09-14

Accept